# Endoscopic Ultrasound-Guided Ablation of Premalignant Pancreatic Cysts and Pancreatic Cancer

**DOI:** 10.3390/diagnostics14050564

**Published:** 2024-03-06

**Authors:** Alejandra Vargas, Priyata Dutta, Eileen S. Carpenter, Jorge D. Machicado

**Affiliations:** 1Department of Medicine, Eastern Virginia Medical School, Norfolk, VA 23510, USA; vargasa@evms.edu; 2Department of Medicine, Trinity Health, Ann Arbor, MI 48197, USA; priyata.dutta@trinity-heatlh.org; 3Division of Gastroenterology and Hepatology, University of Michigan, Ann Arbor, MI 48109, USA; eicarpen@med.umich.edu

**Keywords:** ablation, endoscopic therapy, endoscopic ultrasound, pancreatic cyst, pancreatic cancer

## Abstract

Pancreatic cancer is on the rise and expected to become the second leading cause of cancer-related death by 2030. Up to a one-fifth of pancreatic cancers may arise from mucinous pancreatic cysts, which are frequently present in the general population. Currently, surgical resection is the only curative approach for pancreatic cancer and its cystic precursors. However, only a dismal proportion of patients are eligible for surgery. Therefore, novel treatment approaches to treat pancreatic cancer and precancerous pancreatic cysts are needed. Endoscopic ultrasound (EUS)-guided ablation is an emerging minimally invasive method to treat pancreatic cancer and premalignant pancreatic cysts. Different ablative modalities have been used including alcohol, chemotherapy agents, and radiofrequency ablation. Cumulative data over the past two decades have shown that endoscopic ablation of mucinous pancreatic cysts can lead to cyst resolution in a significant proportion of the treated cysts. Furthermore, novel data are emerging about the ability to endoscopically ablate early and locally advanced pancreatic cancer. In this review, we aim to summarize the available data on the efficacy and safety of the different EUS-ablation modalities for the management of premalignant pancreatic cysts and pancreatic cancer.

## 1. Introduction

Pancreatic cancer constitutes a significant and pressing global health challenge, ranking as the seventh leading cause of cancer-related mortality worldwide and the third in the United States [1]. As of 2023, data from the Surveillance, Epidemiology, and End Results (SEER) program estimated 64,050 new diagnoses and 50,550 deaths attributed to pancreatic cancer in the United States alone [2]. Despite significant progress in understanding the potential risk factors and the availability of advanced diagnostic tools, the incidence of pancreatic cancer continues its ominous ascent, with projections indicating its potential rise to the second leading cause of cancer-related mortality by 2040 [3]. Regrettably, even with advances in pancreatic cancer management and a slight increase in survival rates, the 5-year survival for patients diagnosed with pancreatic cancer still stands at a discouragingly low 12.5% [2]. This poor prognosis has been attributed to several factors, including the aggressive nature of this malignancy, limited treatment success, and late-stage diagnosis.

Pancreatic ductal adenocarcinoma (PDAC), the predominant type of pancreatic cancer, can arise from pancreatic cystic lesions (PCLs) [4]. Mucinous PCLs, which include intraductal papillary mucinous neoplasms (IPMNs) and mucinous cystic neoplasms (MCNs), are the most common types of PCLs and account for approximately 15% of PDAC cases [5]. Therefore, these cystic precursors present a window of opportunity for early detection of PDAC, especially considering that the prevalence of PCLs can be as high as 20% in individuals undergoing cross-sectional imaging [6,7]. Presently, the established approaches for managing PCLs encompass surveillance or surgical resection, each with its limitations and intricacies [8].

Surgery is currently the only potentially route to a curable option for PCLs and PDAC. However, due to increased post procedural risk of complications, morbidity and mortality associated with surgery, many patients are either ineligible for surgery or prefer alternative treatments. In recent years, endoscopic ablative techniques have emerged as a possible alternative treatment modality for PCLs and PDAC. Although the efficacy and safety of these ablative methods are still under investigation, they might play an important role in the multidisciplinary management of premalignant and malignant pancreatic lesions. The goal of this narrative review is to discuss the various endoscopic ablative techniques and the current evidence on their efficacy and safety.

## 2. Clinical Rationale for Endoscopic Ablation Therapy

Endoscopic ultrasound (EUS)-guided ablation offers a less invasive and lower-risk alternative to traditional surgical procedures for PCLs and PDAC [9]. There are a variety of methods used for endoscopic ablation, including ethanol, direct chemotherapy injection, radiofrequency ablation, and photodynamic therapy [10]. Despite the differences in their mechanisms, these endoscopic ablation methods share the common net effect of causing localized necrosis within the lesion while preserving the surrounding healthy pancreatic tissue [11].

Ethanol injection has been used for treating different cystic lesions since the 1980s [12]. Initially used for hepatic and thyroid lesions, this approach has extended its application to various organs [13,14,15,16]. Its safety and efficacy for PCLs were initially evaluated through porcine experiments by Aslanian et al. in 2005 [17], leading to its application in human patients in the same year by Gan et al. [18]. The procedure involves aspirating the cyst volume using a 19- or 22-gauge needle and replacing it with 80–100% ethanol for a 3 to 5 min lavage [18,19,20,21,22,23,24]. However, the variable outcomes and potential side effects of ethanol in ablation of PCLs have prompted to the exploration of alternative treatments [21].

Chemotherapy injection for PCLs was first introduced in 2008 [25]. Agents like Paclitaxel, known for disrupting cell division, and gemcitabine, known for inhibiting DNA synthesis, have shown significant localized tumoricidal effects while minimizing systemic toxicity [25,26,27,28,29]. The direct administration of these agents, either alone or after ethanol lavage, significantly amplifies their localized tumoricidal effect [28]. In an approach without alcohol, the lavage is omitted, and the agent is infused in a volume equal to the aspirated cyst fluid [30]. This technique is also being explored for solid pancreatic lesions, such as PDAC [31].

Radiofrequency ablation (RFA) is applied through a FNA needle using special electrodes (e.g., EUSRA RF) [32]. RFA operates by converting radiofrequency waves into heat (350–500 kHz) to induce thermal coagulation and tissue destruction [33]. This method has also been linked to the induction of T cell-mediated immune responses against tumor antigens, primarily driven by the release of cellular antigens and damage-associated molecular patterns (DAMPs), potentially enhancing antitumor immunity [34,35,36,37].

Photodynamic therapy (PDT) works through an oxygen-dependent reaction between a photosensitizing dye and light that produces localized tissue necrosis. After administration of a photosensitizing agent, the light is delivered in the form of a laser through small optical fibers directly into the tumor under EUS guidance [38,39].

Other ablation techniques such as microwave ablation (MWA), cryoablation, laser ablation (LA), and chemical ablation with Lauromacrogol, can also be administered via EUS. MWA, operating between 900 and 2450 MHz, excels over RFA in heat generation by rapidly oscillating water molecules, achieving deeper and more uniform energy distribution [40]. Its benefits, as shown in a case study on EUS-guided ablation of an unresectable pancreatic mass, include shorter cooling times, better energy transfer efficiency, and reduced susceptibility to heat sinks due to its shorter wavelength [41]. Unlike RFA, MWA is not impeded by tissue impedance, electrical circuit or issues like water vaporization, allowing for deeper penetration and shortening ablation time. Cryothermal ablation, on the other hand, employs a hybrid bipolar probe that synergizes the thermal energy of RFA with the cooling effect of cryogenic gases, inducing cell damage at temperatures below −4 °C [42]. LA, utilizing fine laser fibers, allows for precise targeting of complex, deep-seated pancreatic lesions [43,44]. Lastly, chemical ablation using the sclerosant agent lauromacrogol induces severe local inflammation and intramural fibrosis of vascular structures [45]. However, use of these additional ablation techniques is limited to case reports or small case series, and therefore, will not be described in this review.

## 3. Patient Selection for EUS-Guided Pancreatic Cyst Ablation

Although EUS-guided ablation is recognized as a viable therapeutic option in the management of PCLs, its applicability is still under investigation [9]. The selection of suitable patients requires detailed evaluation of patient’s health and the characteristics of the cyst itself [9,30,46,47,48].

The initial step in evaluating the suitability of PCLs for EUS-guided ablation involves an accurate diagnosis and risk stratification of PCLs [49]. Mucinous cysts without malignant transformation are the types of cysts that most likely benefit of ablation. If the cyst has malignant transformation or is benign (e.g., serous cystadenomas or pseudocysts), endoscopic ablation is unlikely to be beneficial. Given the concern for malignant transformation, the presence of high-grade dysplasia or high-risk features such as main pancreatic duct dilation, the presence of a mural nodule, or common bile duct or pancreatic duct obstruction, are relative contraindications to EUS-guided ablation [50,51,52,53]. Endoscopic ablation is generally favored for unilocular or oligolocular cysts, where the number of locules is limited (up to six), making the ablation process more feasible [29,47,54]. This preference stems from the fact that a single needle pass might not sufficiently deliver therapy to all locules in a cyst with a higher number of septations, potentially leading to treatment failure [54]. The size of the cyst is another important consideration, with those measuring between 2 and 6 cm deemed most suitable for ablation [29,47]. The presence of ductal communication of the cyst with the main pancreatic duct is a relative contraindication for ablation due to concerns about the injectate flowing into the pancreatic duct and causing pancreatitis [46].

Before embarking in endoscopic ablation of PCLs, a surgical evaluation is critical [55]. This assessment should consider the patient’s overall condition, underlying health issues, and personal preferences. Individuals who are deemed non-surgical candidates or who prefer an endoscopic approach rather than surgery or surveillance should be considered for endoscopic ablation. Additionally, life expectancy should be considered in deciding whether the anticipated benefits outweigh the risks of EUS ablation. This can be determined using tools such as the Charlson Comorbidity Index (CCI), the Eastern Cooperative Oncology Group (ECOG) scale, or the Karnofsky Performance Status (KPS) [56,57]. Contraindications for the procedure include pregnancy, irreversible coagulopathy, acute pancreatitis, or the presence of pancreatic necrosis [47] (Table 1).

## 4. Determining PCL Type to Guide Management

Conventional cyst fluid analysis with carcinoembryonic antigen (CEA) level, amylase/glucose concentration, and cytological examination, is useful in the diagnostic work-up of PCLs [58,59,60,61]. However, these tests are limited in their diagnostic accuracy, particularly in terms of sensitivity for identifying malignancy and differentiating PCL subtypes [62]. CEA is useful for differentiating between mucinous and non-mucinous lesions, although its accuracy is only ~70% [58,59]. Cyst fluid glucose has shown better sensitivity (90%) and specificity (82%) than CEA in distinguishing mucinous from non-mucinous PCLs [60,61]. Cytology is highly specific for diagnosing malignancy; however, given that cyst fluid is usually paucicellular or acellular, its sensitivity is of only 54%, and it is rarely positive [63]. As a result, diagnostic inaccuracies are common in clinical practice, leading to unnecessary surgeries or missed early diagnosis of PDAC [64].

In an impetus to improve diagnostic accuracy, new endoscopic diagnostic modalities have been developed, including EUS-guided needle-based confocal laser endomicroscopy (EUS-nCLE), EUS-guided through-the-needle biopsy (EUS-TTNB), and contrast-enhanced harmonic EUS (CE-EUS). EUS-nCLE provides high-resolution microscopic imaging of the cyst lining, enabling real-time assessment of cyst type and degree of dysplasia [65,66]. A recent meta-analysis revealed that nCLE achieves a sensitivity of 85%, a specificity of 99%, and a diagnostic accuracy of over 95% in differentiating cyst types [67]. EUS-TTNB employs a specialized microforceps that can be inserted through a 19-gauge needle to obtain biopsies from the cyst wall or intracystic nodule/mass [68]. CE-EUS is particularly accurate in detecting mural nodules, enhancing their appearance and guiding tissue acquisition [69]. Molecular analysis of cyst fluid using next generation sequencing has also shown high accuracy in diagnosing specific cyst types and to detecting malignancy [70,71,72,73]. A recent network meta-analysis suggests that EUS-nCLE achieves higher diagnostic accuracy in diagnosing PCLs compared to other advanced diagnostic modalities [74].

## 5. Endoscopic Ablation of PCLs

### 5.1. EUS-Guided Ethanol Ablation of PCLs

The first study reporting ethanol ablation of PCLs involved 25 patients and yielded 35% complete cyst resolution (absence of a visible cyst in a 6–12 month follow-up CT scan) and 8.7% partial resolution (over 1 cm or 75% reduction in the original cyst size) [18]. The study used ethanol concentrations varying from 5% to 80% and no complications were observed [18]. Building on these early findings, subsequent studies have predominantly employed ethanol concentrations of 80% or higher and have observed similar rates of complete resolution, adhering to the initial definition of response on follow-up imaging [19,20,21,22,23]. In a double-blind, randomized controlled trial conducted across two tertiary referral hospitals, 42 patients with PCLs were treated with ethanol (*n* = 25) or saline lavage (*n* = 17), with initial saline recipients later crossing over to ethanol [19]. The trial reported a 33% complete cyst resolution with ethanol, assessed 3 to 4 months post-therapy with abdominal CT or MRI. On follow-up imaging conducted at 26 months on 9 out of the original 12 patients who had cyst resolution, there was no cyst recurrence, suggesting a durable response to ethanol [75].

Additional retrospective and prospective studies have focused on refining the technique and identifying cyst characteristics that predict treatment response. For instance, a retrospective study of 13 patients found that two ethanol ablation sessions were more effective than a single session (complete resolution of 38% vs. 0% at 3–12 months) [20]. To evaluate the type of cyst that responds better to alcohol ablation, a prospective study of 91 patients with unilocular or oligolocular PCLs showed a 45% complete resolution at 12 months, with higher rates of resolution for MCNs (50%) than for IPMNs (11%) [23]. Additionally, a retrospective cohort study of 13 patients with MCNs showed 85% complete resolution with ethanol ablation at 6 months [22]. However, the long-term efficacy of alcohol ablation has been questioned by results of a prospective study of 23 patients who had EUS ethanol ablation for PCLs. This study reported that 93.3% of treated cysts persisted and one treated cyst progressed to pancreatic cancer over a 41-month follow-up [21].

Reported adverse events with ethanol ablation include abdominal pain, acute pancreatitis, fever, and intracystic bleeding, in descending order of frequency. In a retrospective study by Choi et al., which involved 214 patients undergoing EUS ethanol ablation of PCLs, 71 individuals (33.2%) experienced adverse events within 30 days following the procedure [24]. The spectrum of adverse events included 22.9% with abdominal pain, 9.8% acute pancreatitis, 0.5% cholangitis, 0.5% bleeding, and 0.9% duodenal stricture. The study also identified that branch duct-IPMNs, multilocular cysts, suspected ethanol leakage during the procedure, and increased cystic fluid viscosity, were independently associated with an increased risk of post-ablation pancreatitis. Additionally, PCLs located in the uncinate process or those with an exophytic portion were more prone to result in post-procedural abdominal pain.

A recent systematic review and meta-analysis of seven studies involving a total of 426 patients, revealed that ethanol ablation of PCLs had a 32% rate of complete cyst resolution (complete disappearance of the cyst in follow-up imaging) and a 36% rate of partial cyst resolution (non-specific percentage reduction in cyst size) [9]. Importantly, the overall risk of adverse events with ethanol ablation was 16% in this systematic review (Table 2).

### 5.2. EUS-Guided Chemoablation of PCLs

The first prospective study to show the efficacy of paclitaxel combined with ethanol was on 14 patients with PCLs and found 79% cyst resolution at 9 months [25]. This was followed by a larger prospective study of the same group among 52 patients with PCLs, which found complete resolution in 55% patients with chemoablation [26]. Another prospective cohort study of 22 patients with PCLs showed that EUS -guided ablation using ethanol and paclitaxel may change the pancreatic cystic fluid characteristics by elimination of baseline DNA mutations, resulting in 75% of image-defined cyst resolution (both complete and partial resolution) [27]. During a median follow up period of 27 months, three patients reported pancreatitis (10%), one patient reported gastric wall cyst, four patients reported abdominal pain (13%), and one patient had peritonitis (3%) [27]. In a large single center prospective study involving 164 patients with PCLs, EUS ablation with ethanol and paclitaxel achieved complete resolution in 114 patients (72.2%) and partial cyst resolution in 31 patients (20%). Moreover, 98.3% of patients with complete resolution remained in remission at 6-year follow-up [29].

However, despite the higher efficacy of adding chemoablation to ethanol lavage, a problem with this combination therapy was that adverse events were still frequent. It was hypothesized that ethanol may be implicated in the pathogenesis of post-procedure pancreatitis and may not increase the efficacy of chemoablation. To test this hypothesis, a randomized controlled trial of 39 patients with mucinous PCLs was conducted (CHARM trial). This trial compared the efficacy on 12-month complete cyst resolution between paclitaxel/gemcitabine with ethanol (*n* = 21) or with saline (*n* = 18) [28]. Both treatment groups had similar cyst resolution in approximately two-thirds of patients. However, the group with ethanol had 28% adverse events (22% abdominal pain, 6% acute pancreatitis), while the group without ethanol had no adverse events [28]. Furthermore, the effect of cyst resolution appears to be durable even up to 3–5 years, with 87% of 23 subjects who achieved complete response at 1 year maintaining complete response 36 months later [76]. In addition, four subjects with no response or partial response at 1 year later had complete cyst resolution. Finally, none of the patients treated with chemoablation developed high grade dysplasia or cancer [76]. Additionally, a recent study of this cohort also demonstrated a reduction in radiologic surveillance and costs with chemoablation as compared to current standard of care [77]. The results of prior studies using chemoablation for PCLs are summarized in Table 3.

While EUS-guided chemoablation has emerged as an alternative for treating PCLs, alcohol-free chemotherapeutic agents’ ablation appears to be effective with less adverse effects. Larger clinical trials are still required to confirm the efficacy of chemoablation in PCLs. In addition, novel particle-engineered forms of chemotherapy agents are being developed and may result in a larger surface area of treatment. In a recent pilot study of 19 subjects with PCLs, large surface area microparticle paclitaxel LSAM-PTX reduced cyst volume in 71% of patients over a 6-month period without major adverse events or systemic absorption [78]. In another small study of six subjects with IPMNs who received 4 doses of LSAM-PTX and who were subsequently followed for up to 32 months, chemoablation was safe and resulted in volume and surface area reduction, morphological changes, and loss of pathogenic mutations [79].

### 5.3. Radiofrequency Ablation (RFA) of PCLs

The data on using RFA for PCLs are more limited compared to intracystic injection of alcohol or chemotherapy. RFA for PCLs was initially studied by Pai et al., demonstrating 100% technical success rate in six patients with PCLs. Among these, complete resolution was achieved in two patients (33%, 2/6) and three patients experienced 50% reduction in cyst size [80]. Another study of 17 patients with PCLs (16 IPMNs and one MCN) reported a 65% (11/17) complete response after EUS-RFA within a year of follow-up, with one patient developing a severe adverse event (jejunal perforation) [81]. In a study of 13 patients with microcystic honeycomb appearance suggestive of SCNs, 62% (8/13) showed a partial response to RFA without any reports of complete response [82]. In a recent prospective single-center study of 12 patients with pancreatic lesions, five with PCLs (four IPMN and one MCN) and seven with pancreatic neuroendocrine tumors (PNETs), EUS-RFA of PCLs demonstrated 60% complete response (3/5), 20% partial response (1/5), and 20% failure (1/5) [83]. Post procedural events did not occur with PCLS. In the PNET group, adverse events included one case of mild acute pancreatitis and two cases of abdominal pain.

## 6. Endoscopic Ablation of Pancreatic Cancer

At present, surgical resection remains the gold standard for achieving long-term survival in patients with PDAC. Unfortunately, only 10–20% of patients are considered surgically resectable [84]. Those with unresectable PDAC typically have limited treatment options, including systemic chemotherapy, chemoradiation or participation in a clinical trial. The lack of other treatment options highlights the need for innovative treatment modalities to improve survival in patients with locally advanced or metastatic PDAC. The role of EUS-guided ablation for PDAC has been investigated for these patients and the data of these studies will be summarized. To our knowledge, there are no data for EUS ablation of resectable PDAC, so this will not be reviewed.

### 6.1. Intratumoral Injection

Intratumoral injection therapies guided by EUS have emerged as a significant area of investigation for their potential efficacy in treating PDAC. These therapies encompass various approaches, including chemotherapy, immunotherapy, viral vector applications, brachytherapy, and gene transfer therapies.

Chemoradiotherapy is a common management strategy for locally advanced PDAC. Despite optimism for newer chemotherapeutics and neoadjuvant therapy for downstaging PDAC, their efficacy in overall survival remains limited for patients with unresectable tumors [85]. One major challenge is the poor penetration of systemic chemotherapy due to tumor desmoplasia [85,86]. Intratumoral chemotherapy or stromal disrupting agents might offer a solution, as they can achieve higher drug concentrations within the tumor compared to systemic administration [85]. This may increase local tumor toxicity, enhance radiation sensitivity, and improve clinical outcomes. However, the effect on survival of local tumor therapy with chemotherapy or immunotherapy agents remains controversial [87].

Recent studies have explored the feasibility, safety, and efficacy of EUS-guided intratumoral injection of chemotherapy. For example, a randomized controlled trial by Li et al. on 18 patients, comparing systemic gemcitabine alone or with tumor implanted 5-fluorouracil capsules, reported no difference on survival or adverse events [88]. In a prospective study of 36 patients with advanced PDAC who underwent EUS-guided fine needle gemcitabine application and then conventional chemotherapy, 97.2% of participants died at 10 months of follow-up [85]. No adverse events were reported with EUS-chemoablation.

EUS-guided intratumoral delivery of immunotherapy, in conjuction with systemic chemoradiotherapy, has shown promising results in locally advanced PDAC. Immunotherapy, designed to reactivate the immune system by counteracting tumor-induced immune suppressors, has been effective using allogeneic lymphocyte cultures and dendritic cells [89,90]. In a phase I-II study of 15 patients with locally advanced PDAC, the combination of intratumoral injection of zoledronate-pulsed dendritic cells, gemcitabine, and αβT cells demostrated encouraging results, with 7 out of 15 patients achieving stable disease according to RECIST criteria [91]. Most of these patients experienced favorable long-term clinical responses, highlighted by a median overall survival of 12 months. Adverse events occurred in one-third of patients, but were only mild, indicating its potential safety and tolerability.

Another emerging locoregional therapy involves oncolytic viruses (OVs), which use engineered viruses to target genes and express genes within the tumor. Experiences with OVs such as TNFerade, Ad5-DS adenovirus, H101 adenovirus, and ONYX-015 have been reported [92,93,94,95,96]. In a prospective study of 50 patients with locally advanced PDAC, TNFerade combined with chemoradiotherapy was administered via EUS-fine needle injection (27 patients) or percutaneous injections (23 patients) [93]. Outcomes included 1 complete response (2%), 3 partial responses (6%), 12 stable diseases (24%) and 19 progressive disease (38%). In a later prospective study of the same cohort, this treatment did not show to significantly prolong survival [92]. In another trial of 21 patients with locally advanced or metastatic PDAC, ONYX-015 showed partial tumor regression in 2 patients, minor response in 2, stable disease in 6, and progressive disease or treatment discontinuation due to toxicity in 11 patients [94]. The treatment was generally safe, with no clinical pancreatitis, although two patients experienced sepsis and two patients had duodenal perforations, which the latter ceased after protocol adjustments. Additionally, a study with nine patients using a combination of intratumoral Ad5-DS and gemcitabine in LAPC patients was safe and well-tolerated [96].

EUS-guided brachytherapy, involving the targeted delivery of radioactive substances such as Iodine-125 and Phosphorus-32, has been studied in locally advanced PDAC. A recent systematic review reported that Iodine-125, evaluated in one retrospective and three prospective studies of 179 patients, is effective in combination with chemotherapy or chemoradiotherapy [97,98,99,100]. Similarly, Phosphorus-32 has shown promise in two prospective studies involving 51 patients [101,102].

Concurrently, gene therapy techniques including BC-819 (in one prospective study with 6 patients), CYL-02 (in one prospective study with 22 patients), and intratumoral RNAi against KRAS (in one prospective study with 15 patients) are being explored for their potential to enhance chemotherapy response [103,104,105]. These EUS-guided interventions represent a significant advancement in pancreatic cancer therapeutics, warranting further research and clinical trials.

### 6.2. Radiofrequency Ablation (RFA)

Recent advances and widespread use of RFA in many solid organ malignancies has enabled this modality to be part of the treatment armamentarium of different types of solid organ malignancies such as hepatocellular carcinoma, breast cancer, and prostate cancer [106]. Its application to the pancreas was first reported in porcine models in 1999 [38]. However, the translation of thermal ablation to pancreatic lesions was limited by the risk of possible complications to the pancreatic parenchyma and surrounding structures.

Use of RFA for PDAC in humans was first applied intraoperatively and then percutaneously under ultrasound guidance. Survival in six studies (four prospective and two retrospective) of patients undergoing RFA for PDAC, ranged from 19 to 25.6 months, with morbidity rate of 0–28% and mortality rate of 0–3% [107]. In five of these studies, RFA was performed during open laparotomy and in one study, RFA was performed via percutaneous approach [107,108,109,110,111,112,113]. Serious adverse events were reported in these studies included severe acute pancreatitis, hepatic failure, portal vein thrombosis, duodenal perforation, biliary injury, hemoperitoneum, gastric ulcer, gastric fistula and gastrointestinal bleeding [107]. When administered percutaneously in patients with locally advanced PDAC, D’Onofrio et al. reported a 93% technical success without any complications [111]. However, the potential limitations of percutaneous RFA include increased body habitus, and overlying abdominal gas as well as risk of excessive radiation exposure while performing CT-guided RFA. Previous studies suggest RFA combined with other systemic approaches provides better survival to patients with PDAC than using RFA alone and that this modality is only effective in non-metastatic disease [108,114].

Emerging data on the use of EUS-RFA for unresectable PDAC have shown promise. EUS-RFA was first reported in 2016 by Song et al. among six patients with unresectable PDAC (four locally advanced and two metastatic) [115]. In this prospective pilot study, technical success was 100% and there were no serious adverse events reported (2 patients had mild abdominal pain) [115]. In a subsequent prospective pilot study by Crino et al., eight patients (seven patients PDAC and one patient with metastatic renal cell carcinoma) received EUS-RFA [116]. EUS-RFA was feasible in all patients, with a mean time of a single RFA application of 58 s and a mean number of applications of 1.5 (range 1–3). On follow-up, there were no major adverse events (only three patients with mild abdominal pain) and imaging showed that 30% of the tumor had been ablated [116]. Similarly, other small single-arm pilot studies have shown the EUS-RFA in unresectable PDAC is feasible in all patients and without major adverse events [117,118]. In another prospective study of 22 patients with PDAC (14 locally advanced and 8 metastatic), a median of 5 RFA sessions were successfully administered and 4% had adverse events (3 had abdominal pain and 1 peritonitis) [119].

In a more recent study of patients with unresectable PDAC, Thosani et al. reported 10 patients who received 22 sessions of EUS-RFA (range 1–4 per patients) concurrently with systemic chemotherapy, achieving 100% technical success, without major adverse events but with post-procedure abdominal pain in 55% of procedures that required no hospital admission [120]. Of 9 patients with imaging follow-up, 7/9 patients had tumor size regression (>50% size reduction in 3) and 2/9 had tumor progression. Moreover, medial survival was 21 months (2/10 patients were alive for over 60 months), which is higher than the expected survival for patients with locally advanced PDAC (9–12%) [120]. The authors stipulate that these good results with RFA are likely explained by: (1) direct treatment of the PDAC through coagulative necrosis; (2) improved chemotherapeutic efficacy as the tumor may become more porous and easier to be penetrated by systemic therapies; and (3) development of systemic antitumor immunity as necrotic particles of the treated PDAC are released into the bloodstream and may be recognized by the immune system [120].

In the only comparative study up to date, Kongkam et al. compared 10 patients who received EUS-RFA + chemotherapy and 12 who received chemotherapy alone [121]. The study found that the group who received RFA had higher rate of tumor necrosis, had slower rate in tumor growth, and required less narcotic utilization. There was no difference on survival. Of the 30 EUS-RFA ablations performed in the RFA cohort, there was one case of mild acute pancreatitis. Otherwise, there was no major adverse events.

The safety of EUS-RFA for pancreatic tumors has been highlighted in a recent systematic review of 13 studies of 134 patients undergoing 165 EUS-RFA procedures for pancreatic indications (40% were neuroendocrine tumors, 29% locally advanced PDAC, 31% other pancreatic lesions) [122]. In this meta-analysis, 10% had post-ablation abdominal, 1% had pancreatitis and <1% had perforation or infection [122]. In a recent large retrospective study of 100 patients who underwent 116 EUS-RFA sessions, close proximity of the pancreatic neoplasm to the MPD (<1 mm proximity) was the only independent predictor for adverse events (OR, 4.10; 95% CI, 1.02–15.22) after pancreatic EUS-RFA [123]. In other studies, reduced temperature, and active cooling during RFA has been found to be associated with fewer complications [38,124,125]. Although temperature is an important parameter to achieve a successful ablation of pancreatic masses, RFA with temperatures higher than 90 degrees Celsius carries a higher risk of thermal injuries such as venous thrombosis, gastrointestinal ulcers, and bleeding [108,126].

### 6.3. Photodynamic Therapy (PDT)

The first clinical study of EUS-PDT in pancreatobiliary malignancies was conducted by Choi (only one had LAPC, while the other three were non-pancreatic), showing no treatment-related complications and stable disease in follow up of 3–7 months [127]. In a case series of 12 patients with locally advanced PDAC, EUS-PDT using photosensitizer porfimer sodium was technically feasible in all patients [128]. However, one-third of patients experienced adverse events related to porfimer sodium including sunburned hands from sun exposure, nausea, photosensitivity, and skin hyperpigmentation. Furthermore, the pharmacology of porfimer sodium necessitates administering the injection 20 to 50 h before the endoscopic session, making it less convenient compared to other endoscopic modalities. EUS-PDT led to tumor necrosis in 50% of patients and facilitated tumor downstaging that permitted surgical resection in 2 patients. In another study Hanada et al., which included eight patients with locally advanced PDAC, EUS-guided PDT caused no major adverse events or photosensitivity, with five patients showing a zone of treatment necrosis at 48 h [129]. However, only one out of eight patients had a survival of 407 days after the procedure.

Since PDT does not use non-ionizing infrared light, it does not carry the risk of accumulating toxicity associated with radiotherapy [39]. PDT treatment requires avoiding skin and eye exposure to direct sunlight or bright indoor light for ~30 days to limit skin necrosis from the photosensitizer. Therefore, a novel photosensitizer, Verteporfin, has been developed, requiring only 7 days of light exposure avoidance [39]. To our knowledge, this photosensitizer has not been used for PDAC. Table 4 summarizes the outcomes of EUS-RFA and EUS-PDT in PDAC [115,116,117,127,129,130].

## 7. Conclusions and Future Directions

In this review, we have provided a comprehensive summary of the current evidence on EUS-guided ablation as a potential treatment for PCLs and PDAC. Although recent research supports the role of EUS-guided ablation for these conditions, there is a clear need for larger, prospective, multicenter studies in more diverse patient populations to firmly establish its safety and efficacy. Future studies should aim to standardize the definition of clinical success, ensuring consistency and comparability of results across different studies. While radiologic resolution is an important objective marker, this is a surrogate endpoint and does not necessarily represent a change in the cyst or tumor biology. Long-term follow-up data are essential to evaluate the durability of the treatment responses and to monitor for any potential lesion recurrence after endoscopic management. Large prospective studies are needed to better understand the safety of these therapies. Finally, further research is needed to refine the selection criteria for individuals with PCLs and PDAC who are most likely to benefit from endoscopic ablation.

## Figures and Tables

**Table 1 diagnostics-14-00564-t001:** Inclusion and Exclusion Criteria for EUS-Guided Pancreatic Cyst Ablation.

Criteria Type	Inclusion Criteria	Exclusion Criteria	
		Absolute	Relative
Patient characteristics	- Not surgical candidates or preference - Reasonable life expectancy	- Pregnancy - Active acute pancreatitis - Pancreatic necrosis present - Irreversible coagulopathy	- Neutropenia - Thrombocytopenia with platelet counts < 50,000
PCL characteristics	- Mucinous cyst (IPMN or MCN) - Unilocular or oligolocular (≤6 locules) - Diameter between 2 and 6 cm	- Confirmed pancreatic cancer - Cytology suspicious for malignancy - Benign PCLs (pseudocyst, serous cystadenoma, lymphangioma, duplication cyst)	- High-grade dysplasia - Cyst with open communication with MPD - High-risk lesions as defined by guidelines

PCL: pancreatic cyst lesion; MPD: main pancreatic duct.

**Table 2 diagnostics-14-00564-t002:** Studies of EUS-Guided Ethanol Ablation for PCLs.

Author, Year	*n*	Ethanol Concentration	Type of Cyst	Outcomes		Adverse Events (%)
				Partial Resolution	Complete Resolution	
Gan 2005 [18]	23	5–80%	IPMN (3), MCN (14), SCN (3), PC (1), Unknown (2)	9%	35%	None
Dewitt 2009 [19]	25	80%	IPMN (10), MCN (10), SCN (3), PC (2)	-	33%	4% pancreatitis, 4% bleeding, 20% abdominal pain
DiMaio 2011 [20]	13	80%	IPMN (13)	-	38%	8% abdominal pain
Caillol 2012 [22]	13	99%	MCN (13)	-	85%	0%
Park 2016 [23]	91	99%	IPMN (9), MCN (12), SCN (33), PC (9), Indeterminate (28)	41%	45%	3% pancreatitis, 20% abdominal pain, 9% fever
Gomez 2016 [21]	23	80%	IPMN (15), MCN (4), non-mucinous (4)	44%	9%	4% pancreatitis, 4% abdominal pain
Choi 2019 [24]	214	-	IPMN (63), MCN (57), SCN (69), PC (25)	44%	25%	10% pancreatitis, 23% abdominal pain, 1% duodenal stricture

IPMN: intraductal papillary mucinous neoplasms; SCN: serous cystic neoplasm; MCN: mucinous cystic neoplasm; SC: serous cystadenoma; PC: pseudocyst.

**Table 3 diagnostics-14-00564-t003:** Studies of EUS-Guided Chemoablation for PCLs.

Author, Year	*n*	Ablative Agents	Type of Cyst	Outcomes		Adverse Events (%)
				Partial Resolution	Complete Resolution	
Oh 2008 [25]	14	Ethanol + Paclitaxel	MCN (2), SCN (3), Others (9)	14%	79%	7% abdominal pain, 7% pancreatitis
Oh 2011 [26]	52	Ethanol + Paclitaxel	MCN (9), SCN (15), Others (28)	14%	67%	2% abdominal pain, 2% pancreatitis, 2% fever
Dewitt 2014 [27]	22	Ethanol + Paclitaxel	IPMN (12), MCN (6), SCN (4)	25%	50%	13% abdominal pain, 10% pancreatitis, 3% peritonitis, 3% gastric wall cyst
Moyer 2017 [28]	39	Paclitaxel/gemcitabine with or without alcohol	IPMN (27), MCN (9), inderterminate (3)	14% without alcohol, 22% with alcohol	67% without alcohol, 61 with alcohol	0% without alcohol vs. 28% with alcohol (22% abdominal pain, 6% pancreatitis)
Choi 2017 [29]	164	Ethanol + Paclitaxel	IPMN (11), MCN (71), SCN (16), Others (66)	20%	72%	4% pancreatitis, 1% pseudocyst, 1% abscess, 4% other
Othman 2022 [78]	19	Large surface area microparticle paclitaxel	IPMN (19), MCN (2)	--	71% decreased in size (35% had >30% reduction)	6% abdominal discomfort, 6% edema, 5% fatigue, 5% headaches
Krishna 2024 [79]	6	Large surface area microparticle paclitaxel	IPMN (6)		74% mean surface area reduction	5% pancreatitis

IPMN: intraductal papillary mucinous and cystic neoplasms; SCN: serous cystic neoplasm; MCN: mucinous cystic neoplasm.

**Table 4 diagnostics-14-00564-t004:** Studies of EUS-RFA and EUS-PDT for Unresectable PDAC.

Author, Year	*n*	Study Groups	Indications	Outcomes	Adverse Events (AEs, %)
Song 2016 [115]	6	EUS-RFA	PDAC	Technical success 100%	No major AEs, mild abdominal pain (33%)
Scopelliti 2018 [117]	10	EUS-RFA	PDAC	Technical success 100%	No major AEs, mild abdominal pain (20%)
Crino 2018 [116]	8	EUS-RFA	PDAC	Technical success 100%, Pancreatic mass reduced 30% at 1 month	No major AEs, mild abdominal pain (38%)
Jiang 2021 [118]	8	EUS-RFA	PDAC	Technical success 100% Pancreatic mass reduced 34% at 1 month	No major AEs
Oh 2022 [119]	22	EUS-RFA	PDAC	Technical success 100% Median survival 24 months	Peritonitis (5%), mild abdominal pain (14%)
Thosani 2022 [120]	10	EUS-RFA	PDAC	Technical success 100% Tumor regression 78% Median survival 21 months	No major AEs, abdominal pain (55%)
Kongkam 2023 [121]	22	EUS-RFA + chemotherapy (10) Chemotherapy alone (12)	PDAC	Higher rate of tumor necrosis, slower rate in tumor growth, and less narcotics in RFA group No difference on survival	No major AEs, mild acute pancreatitis (1 patients with RFA)
Choi 2015 [127]	4 (only 1 PDAC)	EUS-PDT	Locally advanced pancreaticobiliary malignancies (two: Caudate lobe, one: distal bile duct, one: pancreatic tail)	Disease was stable in all patients in follow up of 5 months	No AEs
DeWitt 2017 [128]	12	EUS-PDT	PDAC	Technical success 100%, tumor necrosis in 50% of patients, tumor downstaging that permitted surgical resection in 2 patients	No major AEs, 33% had sunburned hands from sun exposure, nausea, photosensitivity, and skin hyperpigmentation
Hanada 2021 [129]	8	EUS-PDT	PDAC	Treatment necrosis 63% 7 patients had survival duration of 209 days and one patient had survival duration of 407 days	No major AEs Day 2: moderate abdominal pain (13%), minimal pain (53%) Day 14: abdominal pain and diarrhea (12.5), abdominal pain requiring ER visit (12.5), hematochezia (12.5)

PDAC: pancreatic ductal adenocarcinoma; RFA: radiofrequency ablation; PDT: photodynamic therapy; G1: Group 1, G2: Group 2.

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
