# Peer review of "Endoscopic Ultrasound-Guided Ablation of Premalignant Pancreatic Cysts and Pancreatic Cancer"

_diagnostics, 2024, doi:10.3390/diagnostics14050564_

Round 1
Reviewer 1 Report
Comments and Suggestions for Authors
Thank you for giving me the chance to review this manuscript.
This is interesting study, but this is not a rare case and some problems, in the following areas.
1. The author reviews various treatment modalities, but it is difficult to understand the outcome of each. table2 and table4 should include the type of lesions targeted.
Author Response
Dear Editorial Team,
We would like to express our deepest gratitude to the editors and reviewers for their valuable feedback on our manuscript. Below, we provide detailed responses to the specific queries from the reviewers.
Reviewer 1:
Thank you for giving me the chance to review this manuscript.
This is interesting study, but this is not a rare case and some problems, in the following areas.
- The author reviews various treatment modalities, but it is difficult to understand the outcome of each. table2 and table4 should include the type of lesions targeted.
We have summarized all the pertinent the studies and their results. Unfortunately, not every study used the same outcome and given that heterogeneity it is difficult to aggregate results in an easier format. The tables were created with the purpose of highlighting the most important studies and their outcomes, and we hope that this helps readers to understand the study results more easily. As suggested by the reviewer, we have updated Tables 2 and 4 to include the types of lesions targeted by the treatments discussed.
We hope that the revisions made in response to the reviewers' comments satisfactorily address their concerns. Thank you once again for the opportunity to improve our work based on the constructive feedback from the editorial board and the reviewers.
Sincerely,
Alejandra Vargas, Priyata Dutta, Jorge Machicado
Reviewer 2 Report
Comments and Suggestions for Authors
Dear Authors, I had the opportunity to review your interesting paper "Endoscopic Ultrasound Guided Ablation of Premalignant Pancreatic Cysts and Pancreatic Cancer" which I think is a timely contribution to scientifi discussion.
The paper is well written, and the manuscript reflects well what is state in the title
I would suggest one minor revision: pag 2 85-90: when you describe the available needles for rfa you write that eusra needle is an FNA needle. That in not correct in my opinion. EUSRA is an RFA electrode designed for exclusive use in RFA. The ahabib catheter, shici I think is no more on the marke for this purpouse, is a wire with the capability to pass trough aan FNA needle.
best
Author Response
Dear Editorial Team,
We would like to express our deepest gratitude to the editors and reviewers for their valuable feedback on our manuscript. Below, we provide detailed responses to the specific queries from the reviewers.
Reviewer 2:
Dear Authors, I had the opportunity to review your interesting paper "Endoscopic Ultrasound Guided Ablation of Premalignant Pancreatic Cysts and Pancreatic Cancer" which I think is a timely contribution to scientifi discussion.
The paper is well written, and the manuscript reflects well what is state in the title
I would suggest one minor revision: pag 2 85-90: when you describe the available needles for rfa you write that eusra needle is an FNA needle. That in not correct in my opinion. EUSRA is an RFA electrode designed for exclusive use in RFA. The ahabib catheter, shici I think is no more on the marke for this purpouse, is a wire with the capability to pass trough aan FNA needle.
We greatly appreciate the reviewer's suggestion regarding the description of needles for RFA on page 2, lines 85-90. Upon review, we acknowledge the inaccuracy in describing the EUSRA as an FNA needle. As correctly pointed out, EUSRA is indeed an electrode designed exclusively for RFA. We have corrected this error in the manuscript to accurately reflect the current state of FDA-approved devices for EUS-RFA, specifically highlighting the EUSRA RF electrode. We are thankful for this observation, as it has allowed us to improve the accuracy and relevance of our manuscript.
We hope that the revisions made in response to the reviewers' comments satisfactorily address their concerns. Thank you once again for the opportunity to improve our work based on the constructive feedback from the editorial board and the reviewers.
Sincerely,
Alejandra Vargas, Priyata Dutta, Jorge Machicado